# Generation of a Mouse Model Lacking the Non-Homologous End-Joining Factor Mri/Cyren

**DOI:** 10.3390/biom9120798

**Published:** 2019-11-28

**Authors:** Sergio Castañeda-Zegarra, Camilla Huse, Øystein Røsand, Antonio Sarno, Mengtan Xing, Raquel Gago-Fuentes, Qindong Zhang, Amin Alirezaylavasani, Julia Werner, Ping Ji, Nina-Beate Liabakk, Wei Wang, Magnar Bjørås, Valentyn Oksenych

**Affiliations:** 1Department of Clinical and Molecular Medicine (IKOM), Norwegian University of Science and Technology, 7491 Trondheim, Norwaymagnar.bjoras@ntnu.no (M.B.); 2St. Olavs Hospital, Trondheim University Hospital, Clinic of Medicine, Postboks 3250, Sluppen, 7006 Trondheim, Norway; 3Molecular Biotechnology MS programme, Heidelberg University, 69120 Heidelberg, Germany; 4Department of Biosciences and Nutrition (BioNut), Karolinska Institutet, 14183 Huddinge, Sweden

**Keywords:** NHEJ, double-strand breaks, mouse model, lymphocyte, neurodevelopment

## Abstract

Classical non-homologous end joining (NHEJ) is a molecular pathway that detects, processes, and ligates DNA double-strand breaks (DSBs) throughout the cell cycle. Mutations in several NHEJ genes result in neurological abnormalities and immunodeficiency both in humans and mice. The NHEJ pathway is required for V(D)J recombination in developing B and T lymphocytes, and for class switch recombination in mature B cells. The Ku heterodimer formed by Ku70 and Ku80 recognizes DSBs and facilitates the recruitment of accessory factors (e.g., DNA-PKcs, Artemis, Paxx and Mri/Cyren) and downstream core factor subunits X-ray repair cross-complementing group 4 (XRCC4), XRCC4-like factor (XLF), and DNA ligase 4 (Lig4). Accessory factors might be dispensable for the process, depending on the genetic background and DNA lesion type. To determine the physiological role of Mri in DNA repair and development, we introduced a frame-shift mutation in the Mri gene in mice. We then analyzed the development of *Mri*-deficient mice as well as wild type and immunodeficient controls. Mice lacking Mri possessed reduced levels of class switch recombination in B lymphocytes and slow proliferation of neuronal progenitors when compared to wild type littermates. Human cell lines lacking Mri were as sensitive to DSBs as the wild type controls. Overall, we concluded that Mri/Cyren is largely dispensable for DNA repair and mouse development.

## 1. Introduction

Non-homologous end-joining (NHEJ) is a molecular pathway that recognizes, processes, and repairs DNA double-strand breaks (DSBs) throughout the cell cycle [1]. Core NHEJ factors Ku70 and Ku80 form heterodimer (Ku) that is rapidly associated with the DSB sites facilitating recruitment of downstream factors including core x-ray cross-complementing 4 (XRCC4) and DNA ligase 4 (Lig4). XRCC4-like factor (XLF) is also a core factor that binds XRCC4 and stimulates Lig4-dependent DNA ligation. A number of accessory NHEJ factors are required for specific DNA end processing and DNA complex stabilization, in other words, DNA-dependent protein kinase, catalytic subunit (DNA-PKcs), nuclease Artemis and structural components, a paralogue of XRCC4 and XLF (PAXX), and modulator of retroviral infection (Mri) [2,3]. Mice lacking Ku70, Ku80, DNA-PKcs, or Artemis possess severe combined immunodeficient phenotype (SCID), while inactivation of both alleles of the *Xlf* gene results in 2–3-fold reduced B and T cell counts [1,4,5,6,7]. Mice lacking PAXX or Mri possess no or very modest phenotype due to functional redundancy with XLF [8,9,10,11,12]. In contrast, mice lacking either XRCC4 or Lig4 demonstrate p53- and Ku-dependent embryonic lethality, which correlates with massive neuronal apoptosis in the central nervous system [1,13,14,15,16,17]. Combined inactivation of *Xlf* and *Dna-pkcs* results in p53- and Ku70-dependent perinatal lethality in mice [10,18,19]. Moreover, deficiency or haploinsufficiency for *Trp53* rescues synthetic lethality between *Xlf* and *Paxx* [10]. XLF is also functionally redundant in mouse development with Mri [20], recombination activating gene 2, RAG2 [21], and a number of DNA damage response (DDR) factors including Ataxia telangiectasia mutated (ATM) [6], histone H2AX [6,22], mediator of DNA damage checkpoint protein 1 (MDC1) [10], and p53-binding factor (53BP1) [7,23].

Development of B and T lymphocytes depends on programmed DSBs induced by RAG during the V(D)J recombination and NHEJ pathway, which is used for error-prone DNA repair [1]. Moreover, mature B cells replace constant regions of immunoglobulins during the somatic recombination process known as class switch recombination (CSR), when DSBs are initiated by activation-induced cytidine deaminase (AID) and Uridine-*N*-glycosylase (UNG), and NHEJ is used for DNA repair [1,24,25]. Furthermore, the NHEJ process is required for neurodevelopment by preventing neuronal apoptosis [1,26].

*Mri* was initially described as an open reading frame at human chromosome 7 (C7orf49), a factor reversing the resistance to retroviral infection in cell lines [27]. Mri was found to enhance NHEJ [28] and possess an *N*-terminal Ku-binding motif (KBM) [29]. Later, Mri/Cyren was suggested to inhibit NHEJ at telomeres during the S and G2 phases of the cell cycle [30], and finally confirmed to be a bona fide NHEJ factor, which is functionally redundant with XLF in mouse development including the V(D)J recombination and development of the central nervous system [20]. However, it was not clear whether XLF and Mri functionally overlap during the early stages of neurodevelopment (e.g., supporting proliferation and self-renewal of neuronal stem cells). Moreover, due to the lack of a viable mouse model deficient for both XLF and Mri, the impact of Mri on B and T lymphocyte development in vivo is not fully understood.

Here, we introduced a frame-shift mutation to *exon 2* of the murine *Mri* gene. By interbreeding heterozygous parents, we obtained *Mri^−/−^*, *Mri^+/−^*, and *Mri^+/+^* mice at nearly expected ratios. Mri-deficient mice possessed normal body size and number of B and T lymphocytes; however, we detected that stimulated primary mature *Mri^−/−^* B cells had reduced levels of IgG1, and *Mri^−/−^* neurospheres showed a reduced proliferation rate when compared to the *Mri^+/+^* controls.

## 2. Materials and Methods

### 2.1. Mouse Models

All experiments involving mice were performed according to the protocols approved by the Animal Resources Care Facility of Norwegian University of Science and Technology (NTNU, Trondheim, Norway). *Ung^−/−^* mice were described previously [31]. *Mri^+/−^* mice were generated on request and described here for the first time.

### 2.2. Generation of Mri^+/−^ Mice

MRI-deficient (*M^−/−^*) mice were generated through a CRISPR/Cas9 gene-editing approach in 2017 by Horizon Discovery (Saint Louis, MO, USA) upon request from the Oksenych group (IKOM, Faculty of Medicine and Health Science, NTNU, Trondheim, Norway). Single-guide RNA (sgRNA) GGG CTG TCA TCC AAG AGG GGA GG was designed to target *exon 2* of the *Mri* gene in C57BL/6 mice. The 14 bp deletion resulted in a premature stop codon (Figure 1A). Cas9 and sgRNAs were injected into single-cell fertilized embryos, which were then transferred back into pseudopregnant females for gestation. Live-born pups were screened for indel mutation by DNA sequencing. Homozygous pups were used for back-crossing with wild type C57BL/6 mice. Heterozygous *Mri^+/−^* mice were obtained from Horizon Discovery.

### 2.3. Mouse Genotyping

Two polymerase chain reactions (PCRs) were designed to determine the mouse genotypes. The first PCR was performed using TCAGGTCTGCCCTACACTGA and GTGGTGGTGCTTCTCTGTGA primers, detecting both wild type (428 bp) and null (414 bp) alleles (Figure 1B). The second PCR performed with TCAGGTCTGCCCTACACTGA and AGAGGGGAGGACCC primers was used to validate the presence of the WT allele (234 bp, Figure 1B). The PCRs were performed using 50 ng of genomic DNA extracted from murine tissues (e.g., ears, tails), in a final reaction volume of 25 μL, using the Taq 2x Master Mix Kit (New England Biolabs^®^ Inc., Ipswich, MA, USA; #M0270L). A 2.5% agarose gel was used to separate 428 bp and 414 bp PCR products during 18 h at 4 °C, 90 V; and 0.7% agarose gel was used to detect the 234 bp PCR product (75 min, room temperature, 124 V). Genomic DNA isolated from the *Mri^+/+^* and *Mri^−/−^* cells as well as samples with no genomic DNA were used as the PCR controls (Figure 1B).

### 2.4. Fluorescence-Activated Cell Sorting, Splenocyte, and Thymocyte Count

Fluorescence-activated cell sorting (FACS) analysis was performed as previously described [11,32]. Briefly, spleens and thymi were isolated from 2-month-old mice, and splenocytes and thymocytes were counted using Countess™ Automated Cell Counter (Invitrogen, Carlsbad, CA, United States); the cell suspension was spun down and diluted with PBS to obtain a final cell concentration of 2.5 × 10^7^/mL. The samples of 2.5 × 10^6^ splenocytes or thymocytes were blocked for 15 min at room temperature with Mouse BD fragment crystallizable (Fc) Block™ (1:50 dilution) (BD Biosciences, Franklin Lakes, NJ, USA; #553142). The cells were then incubated with fluorochrome-conjugated antibodies (see below) and sorted.

### 2.5. Class Switch Recombination

Class switch recombination (CSR) from IgM to IgG1 was performed as previously described [11]. Naïve B lymphocytes were purified from spleens of 2-month-old mice using EasySep™ mouse B cell enrichment kit (STEMCELL Technology, Vancouver, Canada; #19854), according to the manufacturers’ instructions. For each CSR assay, 2 × 10^4^ cells/200 μL were used in duplicates. The cells were stimulated with LPS (lipopolysaccharides, 40 μg/mL; Sigma Aldrich, St. Louis, MO, USA; #437627-5MG) and IL-4 (Interleukin 4, 20 ng/mL; PeproTech, Stockholm, Sweden; #214-14) for 96 h. Then, the cells were blocked with Fc receptor antibody (2.4G2) and normal mouse serum (Invitrogen, Carlsbad, CA, USA; #10410). The cells were washed in PermWash™ (BD Biosciences, NJ, USA; #554723). Intracellular staining was done using fluorescently tagged anti-mouse antibodies (IgG1-APC) (BioLegend, San Diego, CA, USA; #406610) and the succeeding wash was performed in PermWash. The cells were resuspended in 300 μL of CellFix (BD Biosciences, NJ, USA; #340181). Viable CD19^+^ B lymphocytes were analyzed for IgG1 expression using FlowJo^®^ (Ashland, Oregon, USA) version 7.6 for Windows.

### 2.6. Double Strand Break Sensitivity Assay

The DSBs sensitivity assay was performed as previously described [10,32,33]. Human nearly-haploid HAP1 cells were generated by the Horizon Discovery Group (Waterbeach, Cambridge, UK, #HZGHC005061c001 and #HZGHC005061c004) and are commercially available. HAP1 cells were cultured according to the manufacturer’s instructions. Doxorubicin (Selleckchem, Houston, TX, USA; #S1208), bleomycin (Selleckchem; #S1214), and etoposide (Sigma-Aldrich, St. Louis, MS, USA; #E1383) were used to induce DSBs, and PrestoBlue™ Cell Viability Reagent (Thermo Fisher, Waltham, MA, USA; #A13262) was used to estimate cellular metabolism levels. Briefly, 2000 cells per well were seeded into 96-well plates in 100 μL of Iscove Modified Dulbecco Media (IMDM) medium (day 0). On day 1, 50 μL of the medium was replaced with 50 μL of fresh medium containing doxorubicin, bleomycin, or etoposide, when indicated. Each experimental condition was performed in triplicates. On day 4, 11 μL of 10× PrestoBlue reagent was added to the wells and incubated for 30 min at 37 °C. The cellular viability was estimated according to manufacturer’s instructions, using the excitation/emission wavelengths set at 544/590 nm.

### 2.7. Brain Isolation and Neural Stem Progenitor Cell Culture

The brain was isolated from postnatal day 1 mouse after the cerebellum was removed. The isolated brain was mechanically disrupted in the proliferation medium consisting of Dulbecco Modified Eagle Medium, Nutrient Mixture F12 (DMEM/F12; Thermo Fisher, Waltham, MA, USA; #11330-057), supplemented with penicillin/streptomycin (Thermo Fisher, Waltham, MA, USA; #15140122), B27 without vitamin A (Thermo Fischer Scientific, Waltham, MA, USA; #12587001), EGF (10 ng/mL; PeproTrech, Stockholm, Sweden; #AF-100-15), and bFGF (20 ng/mL; PeproTech; #100-18B). Neural stem progenitor cells (NSPC) form free-floating globular structures referred to as neurospheres. The neurospheres were formed during incubation at 37 °C, 5% CO_2_ and 95% humidity in order to perform the proliferation and self-renewal assay [34].

### 2.8. Neural Stem Progenitor Cell Proliferation and Self-Renewal Assays

Early passage NSPCs (P3–P10) were used throughout all of the NSPC experiments. A PrestoBlue™ Cell Viability Assay was used to investigate the neurosphere proliferation rates, following the manufacturer’s instructions during each incubation on days 1 to 7. The capacity of neural stem cells to maintain their multipotency ex vivo was assessed by determining the number and two-dimensional size of neurospheres [34]. Single NSPCs were plated onto 6-well suspension plates in the proliferation medium on day 0. During days 8 and 10 in culture, images of the entire wells were captured using an EVOS microscope. Only areas between 50 and 1500 pixels were included in the analyses.

### 2.9. Antibodies

The following antibodies were used for FACS. Rat anti-mouse anti-CD16/CD32 (Fc Block, BD Biosciences, San Jose, CA, USA; #553141, 1:50); anti-CD4-PE-Cy7 (Thermo Scientific, Waltham, MA, USA, #25-0042-81, 1:100); anti-CD8-PE-Cy5 (BD Biosciences, San Jose, CA, USA, #553034, 1:100); anti-CD19-PE-Cy7 (Biolegends, San Diego, CA, USA, #115520, 1:100); and hamster anti-mouse anti-CD3-APC (Biolegends, USA, #100312, 1:100).

## 3. Results

### 3.1. Generation of Mri^−/−^ Mice

To investigate the impact of Mri on mouse development, we generated a mouse model with 14 bp frame-shift deletion in *Mri exon 2* on a C57BL/6 background (Figure 1A). Purified sgRNA and Cas9 RNA were introduced to fertilized oocytes, resulting in complete inactivation of the *Mri* gene. *Mri* status (WT, wild type, +/+; heterozygous, +/−; and null, −/−) was confirmed for every experiment by PCR screening (Figure 1B). *Mri^+/+^*, *Mri^+/−^*, and *Mri^−/−^* mice were born from *Mri^+/−^* parents at ratios close to 1:2:1 (Figure 1C). Thirty-day old *Mri^−/−^* mice possessed an average body weight of 15.0 g, which was slightly lower, but not significantly different from the *Mri^+/+^* controls, with a bodyweight of 17.5 g, on average (Figure 1D). The lifespan of *Mri^−/−^* and *Mri^+/−^* mice was monitored for up to 12 months, according to the local regulations. During this time frame, both *Mri^−/−^* and *Mri^+/−^* mice were fertile and had no cancer incidence, similar to the *Mri^+/+^* controls.

### 3.2. Mri^−/−^ Mice Develop Normal Spleens and Thymi

The NHEJ is required for V(D)J recombination in developing B and T lymphocytes, and for CSR in mature B cells [1]. To determine specific functions of Mri in B and T cell development, we first analyzed spleens and thymi isolated from Mri-deficient and WT mice. The average weights of spleens (91 mg) and thymi (69 mg) as well as the average count of splenocytes (121 million) and thymocytes (173 million) was not affected in *Mri^−/−^* mice when compared to *Mri^+/+^* controls (90 mg; 71 mg; 118 million; 186 million, respectively). These numbers were significantly different from the immunodeficient controls, *Dna-pkcs^−/−^* mice (23 mg; 10 mg; six million; five million, respectively) (Figure 1E–H). Moreover, the proportions of CD19^+^ B cells in spleens of six-to eight-week old *Mri^−/−^* mice were on average 60%, which was similar to the proportion of CD19^+^ in *Mri^+/+^* mice (55%, *p* = 0.0668), and significantly higher than the background levels detected in immunodeficient *Dna-pkcs^−/−^* controls (*p* < 0.0001; Figure 2A). The average proportion of CD3^+^ T splenocytes in *Mri^−/−^* mice (21%) was also similar to the one observed in the *Mri^+/+^* controls (22%, *p* = 0.8228), and higher than in the *Dna-pkcs^−/−^* controls (1%, *p* < 0.0001; Figure 2A). *Mri^+/+^* and *Mri^−/−^* mice had similar proportions of CD4^+^ T cells (*p* = 0.8876) and CD8^+^ T cells (*p* = 0.7026) in the spleens, while proportions of CD4^+^ and CD8^+^ T splenocytes in the *Dna-pkcs^−/−^* controls were 4–5-fold reduced (*p* < 0.0001, Figure 2B). In the thymi of six- to eight-week old *Mri^+/+^* and *Mri^−/−^* mice, the proportions of CD4^+^, CD8^+^, and CD4^+^CD8^+^ T cells were similar (*p* > 0.5589), while only background levels were detected in the *Dna-pkcs^−/−^* controls (*p* < 0.0001, Figure 2C).

### 3.3. Class Switch Recombination to IgG1 Is Reduced in Mri^−/−^ Mice

To determine whether Mri deficiency affects CSR, we isolated B cells from the spleens of *Mri^+/+^* and *Mri^−/−^* mice and stimulated the cells with LPS and IL-4. After 96 h, we detected that average IgG1 levels were 33% in *Mri^−/−^* mice, which was significantly lower (*p* = 0.0031) than in the *Mri^+/+^* controls (average 39%; Figure 2D). B lymphocytes isolated from *Ung^−/−^* mice were used as the negative control and possessed on average only 2% of IgG1 at the end of the experiment (96 h), which was lower than in *Mri^+/+^* or *Mri^−/−^* mice (*p* < 0.0001).

### 3.4. Lack of Mri Results in the Reduced Proliferation Rate of Neuronal Stem Progenitor Cells

Previous studies have shown that single knockout of NHEJ DNA repair genes (e.g., *Xrcc4*, *Lig4*, *Ku70*) results in impaired nervous system development in mice [1,13,14]. To determine the impact of Mri on the developing nervous system, we used NSPC isolated from *Mri^+/+^* and *Mri^−/−^* mice at postnatal day 1. We performed four independent experiments using two cell lines from two mice of each genotype. The average proliferation rate of *Mri^−/−^* neurospheres was approximately 35% lower than that in the WT controls, *p* = 0.0043 (Figure 3B).

### 3.5. Normal Self-Renewal Capacity of Mri-Deficient Neuronal Stem Progenitor Cells

To analyze the capacity of NSPCs to maintain the features of stem cells throughout cell divisions and numerous propagations (self-renewal capacity), we counted the number of neurospheres formed in cell culture. In four independent experiments, we plated 10,000 single NSPCs and cultured for eight days. In total, we counted 5123 neurospheres that originated from *Mri^+/+^*, and 4608 from *Mri^−/−^* mice. On average, there were 256 neurospheres in each of the 20 *Mri^+/+^* samples analyzed, and 230 neurospheres in each of the 20 *Mri^−/−^* samples (*p* = 0.7254, n.s., Figure 3C). In addition, images of the neurospheres were collected and the surface was calculated using *ImageJ* software. Inactivation of *Mri* did not affect the average diameter of neurospheres, which was 461 px^2^ on average in *Mri^+/+^* and 427 px^2^ in *Mri^−/−^* neurospheres, *p* = 0.4915 (Figure 3D). We concluded that Mri is dispensable for the self-renewal capacity of NSPC.

### 3.6. Human HAP1 Cells Lacking Mri Possess Normal Levels of Sensitivity to DNA Double-Strand Breaks

Deficiency for XRCC4, LIG4, XLF, or DNA-PKcs results in hypersensitivity to DSBs in human HAP1 cells [10,32,33]. To determine the effect of Mri on DSB sensitivity, we obtained two independent clones of *MRI*-deficient HAP1 cells as well as WT and *XRCC4*-deficient controls. We exposed the HAP1 cells to DSB-inducing agents bleomycin (0 to 0.4 μM), doxorubicin (0 to 4 nM), and etoposide (0 to 160 nM), and evaluated the cell viability four days later (Figure 4). We observed no hypersensitivity of HAP1 cells lacking Mri when compared to WT controls. However, cells lacking XRCC4 were hypersensitive to all indicated compounds, bleomycin, doxorubicin, and etoposide (*p* < 0.0001, Figure 4).

## 4. Discussion

We have generated a new knockout mouse model with 14 bp deletion in *exon 2* of the *Mri* gene, *Mri^−/−^*. While we were characterizing our mouse model, another group reported an independently-generated *Mri*-deficient mouse [20], which possessed a similar phenotype. Thus, our observations are confirmatory to the findings observed by the Sleckman group [20].

The mice lacking Mri were live-born at expected ratios and demonstrated normal growth and development of lymphoid organs. *Mri^−/−^*, *Mri^+/−^*, and *Mri^+/+^* mice possessed similar sizes of spleens and thymi, a similar number of splenocytes and thymocytes, and proportions of B and T cells (Figure 1). Similar to Mri-deficient mice, *Paxx^−/−^* mice did not have a detectable phenotype [8,9,10,11,12]. However, inactivation of other NHEJ factors resulted in a reduced number of B and T cells (*Xlf^−/−^* mice, [4,5,6,7,18,21,23]), and block in B and T cell development (e.g., *Artemis^−/−^* [35], *Dna-pkcs^−/−^* [36], *Ku70^-/-^* [37], *Ku80^−/−^* [38]; or even embryonic lethality in *Xrcc4^−/−^* [39] and *Lig4^−/−^* [40]).

The CSR to IgG1 was reduced in primary B cells isolated from *Mri^−/−^* mice when compared to WT controls (Figure 2), which suggests that Mri is involved in specific DNA repair-mediated event. Furthermore, we isolated neuronal stem progenitor cells from *Mri^−/−^* brains and found that these cells proliferate slower when compared to *Mri^+/+^* controls. Reduced proliferation rates of Mri-deficient neuronal stem progenitor cells could be explained, as one option, by lower expression or lack of factors functionally redundant with Mri in these cell types. Future studies would be required to address this option. Moreover, future studies may address questions such as neurological defects and cognitive functions in mice lacking Mri as well as whether the Mri-deficient mice are prone to infections.

In addition, we found that human nearly haploid HAP1 cell lines lacking Mri possessed no proliferation defect or hypersensitivity to DNA damaging agents such as etoposide, doxorubicin, and bleomycin (Figure 4). Previously, it was reported that murine embryonic fibroblasts (MEF) lacking Mri were hypersensitive to ionizing radiation when compared to WT controls, although the sensitivity was less obvious than in XLF-deficient MEFs [20]. The discrepancy between our and previously published data could be due to the usage of different cell types, the difference between species as well as distinct ways to induce DNA damages (e.g., chemicals vs. irradiation). Further studies involving various cell type models originated from different species, and using various DNA damaging strategies would deepen our understanding of the specific functions of Mri in DNA repair in mammalian cells. Overall, we concluded that the lack of Mri has a rather minor effect on murine and human cells.

Combined inactivation of *Mri* and *Xlf* [20], however, revealed an important role of Mri in mouse development, which was overlooked due to its functional redundancy with XLF. Synthetic lethality between *Mri* and *Xlf* complicated studies of genetic interaction between these factors in vivo. There are several potential ways to overcome this challenge. One option is to use conditional knockouts of *Xlf* or *Mri* genes. Moreover, there might be another genetic-based approach. Inactivation of one or two alleles of *Trp53* partially rescued synthetic lethality between *Xlf* and *Dna-pkcs* [10,18,19] and *Xlf* and *Paxx* [10]. In addition, deficiency or haploinsufficiency for *Trp53* rescued embryonic lethality of *Lig4^−/−^* and *Xrcc4^−/−^* mice [13,14]. Inactivation of the *Atm* gene rescued embryonic lethality of *Lig4^−/−^* mice [41]. Finally, inactivation of both alleles of *Ku80* rescued embryonic lethality of *Lig4^−/−^* mice [17], and inactivation of *Ku70* rescued synthetic lethality between *Xlf* and *Dna-pkcs* [19]. Based on these data, one can speculate that inactivation of *Trp53*, *Atm*, *Ku70*, or *Ku80* will rescue synthetic lethality between *Xlf* and *Mri*. Moreover, given the critical roles of Ku70 and Ku80 in the initiation of classical NHEJ, one could propose that mice lacking all known NHEJ factors (e.g., *Ku70^−/−^Ku80^−/−^Dna-pkcs^−/−^Artemis^−/−^Xlf^−/−^Paxx^−/−^Mri^−/−^Xrcc4^−/−^Lig4^−/−^*) will be viable, indistinguishable from Ku-deficient mice, and serve as a suitable in vivo model to investigate alternative end-joining, A-EJ.

## 5. Conclusions

A new *Mri*-deficient mouse model was generated. *Mri*-deficient mice possessed normal body size and number of B and T lymphocytes; however, Mri is required for an efficient class switch recombination process in mature B cells. *Mri^−/−^* neurospheres showed a reduced proliferation rate, but similar self-renewal capacity when compared to the *Mri^+/+^* controls.

## Figures and Tables

**Figure 1 biomolecules-09-00798-f001:**
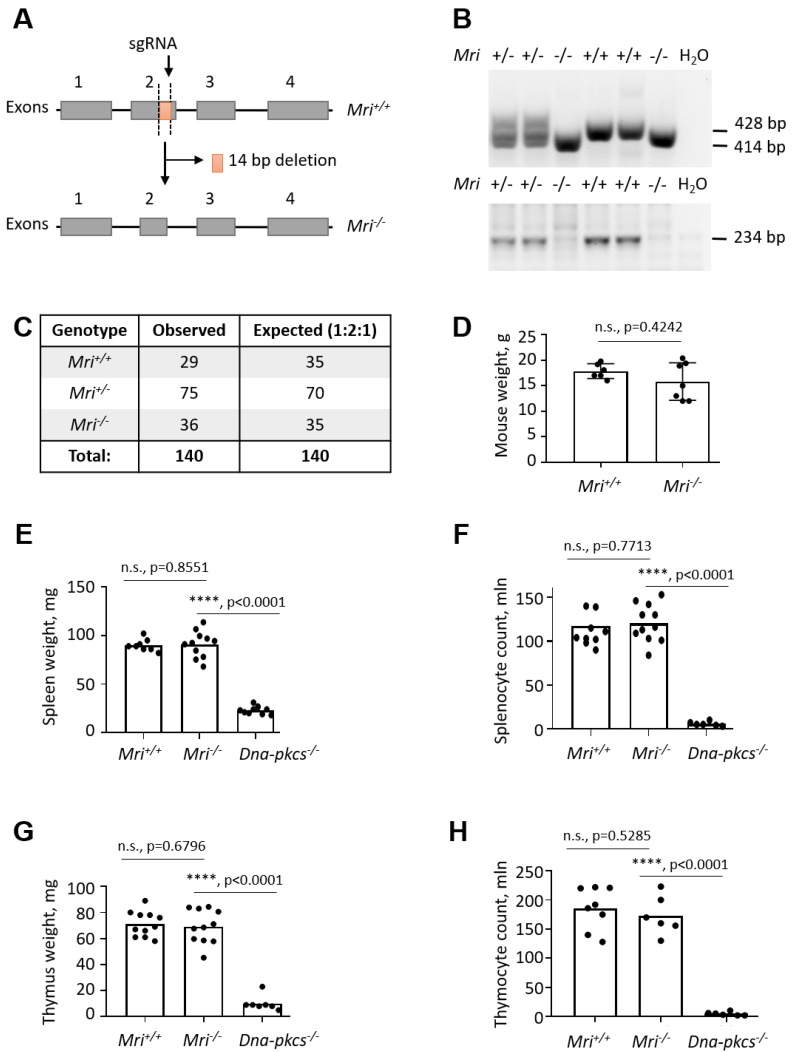
Generation of *Modulator of retrovirus infection^−/−^* (*Mri^−/−^*) mice. (**A**) Top. Schematic diagram of murine *Mri* locus indicating the frame-shift mutation in the *exon 2*, induced by the single guide RNA (sgRNA) and resulting in a 14 bp deletion. (Bottom) Resulting *Mri^−/−^* locus lacking part of the *exon 2*. (**B**) Top. Polymerase chain reaction (PCR)-based genotyping strategy reveals the *Mri* WT allele (428 bp) and *Mri* null allele (414 bp). (Bottom) WT gene validation PCR revealed the *Mri* wild type allele (234 bp). (**C**) Analyses of 140 pups born from *Mri^+/-^* parents revealed expected genetic distribution of *Mri^+/+^* (29), *Mri^+/−^* (75), and *Mri^−/−^* (36) mice, which is close to the Mendelian distribution 1:2:1. (**D**) Body weight of six to eight week old *Mri^+/+^* mice (n = 6) was not distinguishable from *Mri^−/−^* mice (n = 7), *p* = 0.4242. (**E**) The weight of spleens isolated from *Mri^+/+^* (n = 8) and *Mri^−/−^* mice (n = 11) was not significantly different, *p* = 0.8551. Spleen size in immunodeficient *Dna-pkcs^−/−^* mice (n = 10) was reduced when compared to the *Mri^+/+^* and *Mri^−/−^* mice, *p* < 0.0001. (**F**) Splenocyte count was not affected in the *Mri^−/−^* mice (n = 11) when compared to the *Mri^+/+^*(n = 10), *p* = 0.7713. A number of splenocytes in immunodeficient *Dna-pkcs^−/−^* mice (n = 6) was significantly reduced when compared to *Mri^+/+^*and *Mri^−/−^* mice, *p* < 0.0001. (**G**) The weight of thymus from *Mri^+/+^*(n = 11) and *Mri^−/−^* (n = 11) mice was similar, *p* = 0.6796. Thymus size in immunodeficient *Dna-pkcs^−/−^* mice (n = 7) was significantly reduced when compared to *Mri^+/+^*and *Mri^−/−^* mice, *p* < 0.0001. (**H**) The thymocyte count was nearly identical in *Mri^+/+^*(n = 8) and *Mri^−/−^* (n = 6) mice, *p* = 0.5285. A number of thymocytes in immunodeficient *Dna-pkcs^−/−^* mice (n = 6) was significantly reduced when compared to *Mri^+/+^*and *Mri^−/−^* mice, *p* < 0.0001.

**Figure 2 biomolecules-09-00798-f002:**
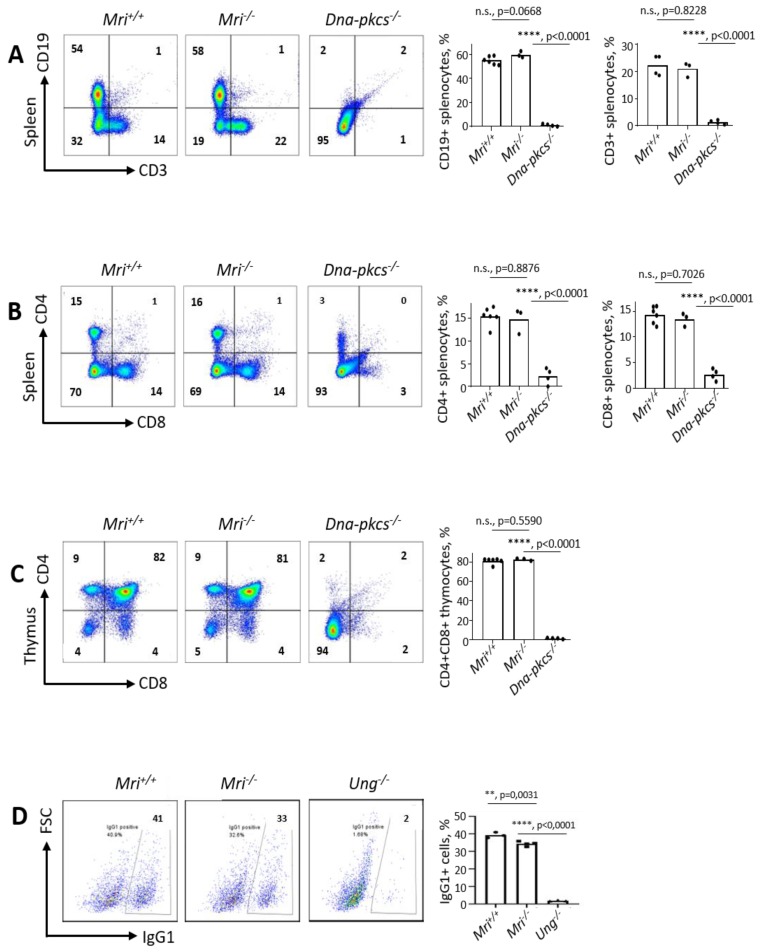
Lymphocyte development in *Mri^−/−^* mice. (**A**) Proportions of T (CD3^+^) and B (CD19^+^) cells in the spleens of *Mri^+/+^*(n = 6), *Mri^−/−^* (n = 3), and *Dna-pkcs^-/-^* (n = 4) mice. Proportions of T and B cells were similar in *Mri^+/+^*and *Mri^−/−^* mice, *p* > 0.0667, and were only background levels in immunodeficient *Dna-pkcs^−/−^* mice, *p* < 0.0001. (**B**) Proportions of CD4^+^ and CD8^+^ T splenocytes in *Mri^+/+^*(n = 6), *Mri^−/−^* (n = 3), and *Dna-pkcs^−/−^* (n = 4) mice. *Mri^−/−^* mice possessed similar proportions of CD4^+^ T helper and CD8^+^ T cytotoxic cells when compared to *Mri^+/+^* mice, *p* = 0.8876 and *p* = 0.7026, respectively. Only background levels of CD4^+^ and CD8^+^ T cells are present in immunodeficient *Dna-pkcs^−/−^* spleens, *p* < 0.0001. (**C**) Proportions of CD4^+^, CD8^+^, and CD4^+^CD8^+^ thymocytes in *Mri^+/+^*(n = 6), *Mri^−/−^* (n = 3), and *Dna-pkcs^−/−^* (n = 4) mice. Proportions of T cell types in *Mri^−/−^* mice were similar to the ones detected in *Mri^+/+^*mice, *p* > 0.5589, and higher than in *Dna-pkcs^−/−^* mice, *p* < 0.0001. (**D**) CSR to IgG1 in primary B splenocytes isolated from the *Mri^−/−^* mice (n = 4) was reduced when compared to the cells from the *Mri^+/+^*mice (n = 3), *p* = 0.0032. CSR to IgG1 was significantly reduced in the *Ung^−/−^* B cells (n = 3) when compared to the *Mri^+/+^* and *Mri^−/−^*, *p* < 0.0001.

**Figure 3 biomolecules-09-00798-f003:**
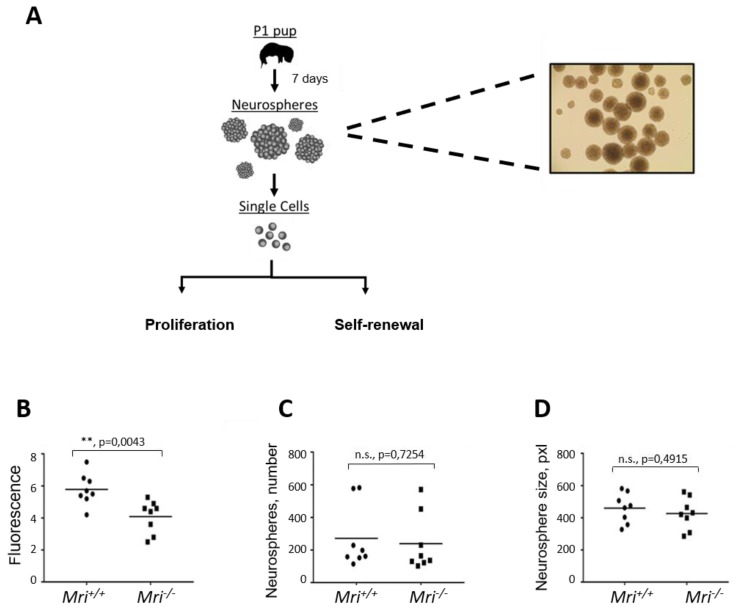
Characterization of neurogenesis in *Mri^−/−^* mice. For each experiment, four independent cell lines isolated from two mice were used (n = 8). (**A**) Neurosphere isolation diagram from *Mri^+/+^* and *Mri^−/−^* mice at postnatal day 1. (**B**) Neurosphere proliferation isolated from the *Mri^−/−^* mice was reduced when compared to the *Mri^+/+^* mice, *p* = 0.0043. (**C**) Number of neurospheres formed in cell culture for eight days. *Mri^−/−^* and *Mri^+/+^* neurospheres possessed similar self-renewal capacity, *p* = 0.7254. (**D**) Neurosphere size isolated from *Mri^−/−^* and *Mri^+/+^* mice were similar, *p* = 0.4915. The surface of neurospheres, pxl. Areas between 50 and 1500 pixels were included in the analyses. Four independent experiments using two cell lines of each genotype were performed in all experiments (**A**–**C**). *p* values were calculated using the unpaired *t*-test. The horizontal bars represent the average.

**Figure 4 biomolecules-09-00798-f004:**
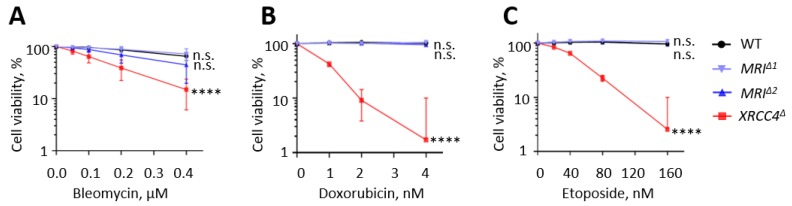
Sensitivity to DSBs in *Mri-*deficient HAP1 cells. Sensitization of WT, two independent Mri-deficient clones, *MRI^∆1^* and *MRI^∆2^*, and *XRCC4^∆^*HAP1 cells to bleomycin (**A**), doxorubicin (**B**), and etoposide (**C**) at indicated concentrations. Results are from the mean (SD) of three repeats. Cell viability (%) represents the relative proportion of the fluorescence normalized to untreated cells. Comparisons between every two groups were made using one-way ANOVA, GraphPad Prism 8. WT, *MRI^∆1^*, and *MRI^∆2^* vs. *XRCC4^∆^*, *p* < 0.0001 (****); WT vs. *MRI^∆1^*, *p* = 0.9983 (n.s); WT vs. *MRI^∆2^*, *p* = 0.1295 (n.s); *MRI^∆1^* vs. *MRI^∆2^*, *p* = 0.1791 (n.s).

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
