# Peer review of "Generation of a Mouse Model Lacking the Non-Homologous End-Joining Factor Mri/Cyren"

_biomolecules, 2019, doi:10.3390/biom9120798_

Round 1
Reviewer 1 Report
Castañeda-Zegarra et al. generated a mouse model lacking the non-homologous end-joining factor Mri/Cyren. They introduced a frame-shift mutation in the Mri gene in mice and investigated the development of Mri-deficient mice in comparison to wild type and immunodeficient controls. Mice lacking Mri demonstrated reduced levels of class switch recombination in B lymphocytes and slower proliferation of neuronal progenitors when compared to wild type mice. Further, human cell lines lacking Mri were as sensitive to DSBs as WT controls. Based on their results they concluded that Mri/Cyren is largely dispensable for DNA repair and mouse development.
The experiments are straightforward and well designed. The Mri-deficient mouse model improves our understanding in vivo about the non-homologous end-joining accessory factor Mri/Cyren.
Minor comments
Is there anything known about the life span of Mri-/- mice?
Are Mri-/- mice prone to infections?
Show Mri-/- mice any neurological deficits?
Author Response
We are grateful to Reviewer 1 for the comments.
The lifespan of Mri-/- and Mri+/- mice was monitored for up to 12 months, according to the local regulations (Comparative Medicine Core, NTNU, Trondheim, Norway). During this time frame, both Mri-/- and Mri+/- mice were fertile and had no cancer incidence, similarly to the Mri+/+ controls. At least ten mice of each genotype were monitored. (Lines 175-177 of the updated manuscript, Results section, 3.1.)
We have no data yet on whether Mri-/- mice are prone to infections. We kept all the mice in the germ-free unit together with immunodeficient mice. This point can be addressed in future studies (lines 297-299, Discussion).
We did not observe any neurological deficits in Mri-/- mice. This point will be of interest for future studies, including aspects of cognitive functions. We added these statements to the Discussion section (lines 297-299, Discussion).
Reviewer 2 Report
The manuscript by Castañeda-Zegarra et al describes the generation and characterization of the mouse model of Mri deficiency. The authors did in-depth analyses of the mouse model and found that genetic ablation of Mri is largely dispensable for DNA double strand break repair and mouse development and that Mri knockout reduces levels of class switch recombination as well as proliferation rates of neuronal stem progenitor cells. The content of the text clearly justifies the title and the abstract and the data are of high quality. The authors should consider addressing the following concerns:
There is little new information in this work, since the Mri-deficient mice were already reported by Hung et al, 2018, Mol Cell, 71, 332–342. However, it is good to see reproducible results of a mouse model generated in different ways by two independent groups. The authors should clearly state in the discussion that their work is confirmatory to the findings observed by the Sleckman group on Mri-deficient mice. The authors should provide details on the number of mice used for each experiment in Figures 1-3. They only provided this information for Figure 1C. The authors should provide possible explanations as to why Mri loss results in reduced proliferation of neuronal stem progenitor cells. It is interesting that Mri knockout HAP1 cells display no hypersensitivity to DSB-inducing agents, whereas Hung et al showed that Mri deficient MEFs are sensitive to ionizing radiation. The authors should address the discrepancy between these results. The authors should consider testing DSB inducing agents in Mri deficient MEFs or additional human cell lines?Author Response
We are grateful to Reviewer 2 for the constructive comments and suggestions.
We added the statement to the Discussion that our work is confirmatory to the findings observed by the Sleckman group (lines 282-283, Discussion)
The number of mice used in Figures 1-2 is now indicated in Figure legends as (“n”).
Figure 1D. Mri+/+ mice (n=6); Mri-/- mice (n=7)
Figure 1E. Mri+/+ (n=8), Mri-/- (n=11), Dna-pkcs-/- (n=10)
Figure 1F. Mri+/+ (n=10), Mri-/- (n=11), Dna-pkcs-/- (n=6)
Figure 1G. Mri+/+ (n=11), Mri-/- (n=11), Dna-pkcs-/- (n=7)
Figure 1H. Mri+/+ (n=8), Mri-/- (n=6), Dna-pkcs-/- (n=6)
Figure 2A. Mri+/+ (n=6), Mri-/- (n=3), Dna-pkcs-/- (n=4)
Figure 2B. Mri+/+ (n=6), Mri-/- (n=3), Dna-pkcs-/- (n=4)
Figure 2C. Mri+/+ (n=6), Mri-/- (n=3), Dna-pkcs-/- (n=4)
Figure 2D. Mri+/+ (n=3), Mri-/- (n=4), Ung-/- (n=3)
For the Figure 3, we used two mice of each genotype, and four independent cell lines were analyzed from each mouse. Therefore, n=8 means number of cell lines.
Figure 3. Mri+/+ (n=8); Mri-/- (n=8).
Reduced proliferation rates of Mri-deficient neuronal stem progenitor cells could be explained, as one option, by lower expression or lack of factors functionally redundant with Mri in this cell type. Future studies would be required to address this option (lines 294-297, Discussion).
Previously, it was reported that murine embryonic fibroblasts (MEF) lacking Mri are hypersensitive to ionizing radiation when compared to WT controls, although the sensitivity is less obvious than in XLF-deficient MEFs [21]. The discrepancy between our and previously published data could be due to the usage of different cell types, the difference between species, as well as distinct ways to induce DNA damages (e.g., chemicals vs irradiation). Further studies involving various cell type models originated from different species, and using various DNA damaging strategies, would deepen our understanding of the specific functions of Mri in DNA repair in mammalian cells (lines 304-309, Discussion).